# On the Microstructure and Mechanical Properties of CrN_x_/Ag Multilayer Films Prepared by Magnetron Sputtering

**DOI:** 10.3390/ma13061316

**Published:** 2020-03-13

**Authors:** Chunfu Hong, Ping He, Jun Tian, Fa Chang, Jianbo Wu, Pingshan Zhang, Pinqiang Dai

**Affiliations:** 1Fujian Provincial Key Laboratory of Advanced Materials Processing and Application, Fujian University of Technology, 3 Xueyuan Road, University Town, Fuzhou City, Fujian 350118, China; heping2629@163.com (P.H.); tianj2003@126.com (J.T.); cfzyzs_2011@163.com (F.C.); 2School of pharmaceutical chemical and material engineering, Taizhou University, 1139 Shifu Road, Taizhou City, Zhejiang 318000, China; wujb@tzc.edu.cn; 3Huamin nanping automobile fittings group Co. LTD., Changsha Industrial Park, Nanping City, Fujian 353000, China; pingshanzhang@163.com

**Keywords:** CrN_x_/Ag, multilayer, magnetron sputtering, hardness, solid lubrication

## Abstract

CrN_x_/Ag multilayer coatings and a comparative CrN_x_ single layer were deposited via reactive magnetron sputtering. In multilayer coatings, the thickness of each CrN_x_ layer was constant at 60 nm, while that of the Ag layer was adjusted from 3 to 10 nm. Microstructure of the films was characterized by X-ray diffraction and transmission electron spectroscopy. The results suggest that the film containing 3 nm of Ag layer presents a nanocomposite structure comprising fine nano-grains and quasi-amorphous clusters. With Ag layer thickness reaching 4.5 nm and above, Ag grains coalesce to produce continuous an Ag layer and exhibit (111) preferential crystallization. Hardness of the films was detected by nanoindentation and it reveals that with increasing the Ag layer thickness, the hardness continuously decreases from 30.2 to 11.6 GPa. Wear performance of the films was examined by the ball-on-disk test at 500 °C. The result suggests that the out-diffusion of Ag towards film surface contributes to the friction reduction, while the wear performance of films depends on the thickness of the Ag layer.

## 1. Introduction

In recent years, with increasing demands of low friction contact or dry machining at elevated temperature, coatings combining phases of hard nitride and solid lubrication have been extensively studied [1,2,3,4,5]. Ag is a soft metal that provides low friction against hard surfaces at temperatures ranging from room temperature (RT) to 700 °C, and thus, becomes a good self-lubricating component combining with the nitride phase. For instance, S. M. Aouadi et al. studied the mechanical properties of ZrN/Ag nanocomposite films prepared by co-sputtering of Zr and Ag targets, and they found the films with 6~12 at. % Ag provided lower wear rate than monolithic ZrN during RT sliding [6]. H. Köstenbauer et al. reported that magnetron sputtered TiN/Ag nanocomposite films possessed lower friction coefficient than monolithic TiN. They suggested a critical Ag content was required to activate the segregation of Ag towards the film surface and thus, to reduce the friction coefficient and wear loss at higher temperature [7]. C. P. Mulligan et al. also reported the friction and wear behavior of CrN/Ag nanocomposite films with regard to the film content, microstructure and temperature effects [8,9,10,11,12]. They suggest that higher Ag content results in a lower friction coefficient, while excessive Ag content decreases the hardness of films sharply and over-accumulates on the films surface, thus reducing the wear resistance of the film surface. 

The microstructure of the film is the key factor determining the Ag diffusion rate and thus property conservation under the varying service temperature. It is still under research how to design the composition of nitride/Ag nanostructure films that fulfill the temperature-adaptive friction and wear performances. C. P. Mulligan et al. reported that a compact CrN cap layer acting as a diffusion barrier prevented surface accumulation of Ag, thus the broken CrN layer along the wear track became the only opening for Ag diffusion and the wear resistance of CrN/Ag was consequently enhanced [13]. For films with combined components and properties, multilayer is a good candidate. There have been a lot of researches and applications of nitride/nitride, nitride/metal multilayer films concerning improved hardness, toughness, solid lubrication, wear resistance, and so on [14,15,16,17,18,19]. Y. B. Zhu et al. reported the sluggish out diffusion of Ag in TiSiN/Ag multilayer coatings by ion plating, and they suggested that Ag diffuse to the film surface through the “micro-channels” in the TiSiN matrix [5]. M. Baraket et al. prepared CrN/Ag and CrSiN/Ag multilayer films at constant Ag layer thickness of 4 nm and nitride layers of 4~20 nm. They found CrN(20 nm)/Ag(4 nm) coating presented the best wear resistance [20]. S. H. Yao et al. reported superior adhesion and corner wear properties of CrN/Ag multilayer comparing with the CrN single layer [21]. Nitride/Ag multilayer could be a good choice for surface protection especially at elevated temperature, however, detailed studies on Ag microstructure and diffusion process are still few.

CrN_x_ films possess excellent toughness, hardness, wear resistance, and a beneficial deposition rate, but have widely been prepared as protective coatings [22,23,24]. Ways to reduce the friction coefficient will meet extended applications and benefits. The aim of this paper is to study the relationships between the microstructure and mechanical properties in CrN_x_/Ag multilayer films. It is not only the influence of the volume content of Ag (tuned by layer thickness), but also the microstructure evolution process will be investigated. For better studying the out-diffusion behavior of Ag, films were prepared with varied Ag layer thicknesses, while CrN_x_ was fixed at a larger layer thickness.

## 2. Materials and Method

### 2.1. Deposition Tool and Substrate 

CrN_x_ single layer and CrN_x_/Ag multilayer films were deposited by reactive magnetron sputtering in Ar/N_2_ atmosphere. The home-made deposition system was installed with four independent power targets and a three-axis substrate rotation rig to produce uniform films on flat substrates. The targets are Ø 60 × 5 mm in size and normally 70 mm from the substrate surface. A resist heater is equipped on the substrate holder to sustain a stable substrate temperature below 800 °C. The substrate used is a square and a single side polished Si (100) wafer that is 3.6 × 3.6 cm^2^ in size. Films are deposited on the polished side, and each film is deposited twice on two pieces of substrates. 

### 2.2. Process of Films Preparation

The multilayer films were prepared by alternatively sputtering Cr (99.95%) and Ag (99.99%) targets. After attaining a base vacuum pressure below 2 E–3 Pa, the chamber was backfilled with Ar (99.995% purity) to 0.4 Pa, and then the substrates were heated to 350 °C at a rate of 10 °C/min. Prior to deposition, the substrate was -500 V biased and discharge-cleaned in Cr glow (at 0.45 W/cm^2^) for 30 min. The deposition process can be described in the following sections: (i) by decreasing the bias to -35 V and increasing Cr target power density to 3.2 W/cm^2^, Cr adhesion layer was deposited for 5 min; (ii) by introducing and gradually increasing N_2_ (99.999%) stream to a partial pressure of 0.08 Pa within 20 min, a Cr-N compositional transition layer was deposited; (iii) the CrN_x_/Ag multilayer was deposited by alternately depositing CrN_x_ and Ag layers.

In each bilayer period, the durations for CrN_x_ and Ag deposition were 50 s and 5 s, respectively. The power density of the Cr target was kept at 3.2 W/cm^2^, while the Ag target power was varied as shown in Table 1. CrN_x_ single layer was also prepared at the same conditions except for Ag power.

### 2.3. Characterization of Films Morphology

The film thickness was detected by measuring a fall between the film surface and adjacent uncoated substrate, using a stylus profilometer (Dektak 8, Veeco, America). The fall was made through a mask of an iron plate attached to a screw that fixed the substrate to the rotation rig. At least two data points were received for each film to calculate the thickness.

The morphology of the film surface and cross section was characterized by a field emission scanning electron microscope (FE-SEM, Nova Nano SEM 450, FEI, Hillsboro, America).

### 2.4. Characterization of Composition and Microstructure

The film composition was characterized through an energy dispersive spectroscopy (EDS, E-max Energy EX-350, Horiba, Japan), equipped on SEM.

The crystallographic structure of the coatings was assessed by an X-ray diffractometer (XRD, D/max-rA, Rigaku, Japan), performing in a θ-2θ geometry with a step size of 0.02° and scan speed at 4°/min, using Cu Kα X-radiation. The result was calibrated using full annealed Silicon-640; peak-decomposition was carried out using the Pearson type VII function, and grain size of films was calculated by Scherrer formula [25,26].

Microstructure investigation was carried out through a high-resolution transmission electron microscope (TEM, JEM 2010, JEOL, Japan), operating at 200 kV. In multilayer films, the thicknesses of CrN_x_ and Ag layers were also characterized by TEM study.

### 2.5. Characterization of Hardness

The hardness of the films was examined on a depth sensing nanoindenter (G200, Agilent, America) with a standard Berkovich indentation tip. The indentation was performed using Dynamic Contact Module so that the loading process was controlled by a constant drift rate of 0.5 nm/s and a constant strain rate of 0.05/s. The indentation depth of all the films ranged from 390 to 400 nm, about 1/10 of the film thickness. The hardness (*H*) and elastic modulus (*E*) were measured based on the Oliver and Pharr method with a hypothetic Possion’s ratio of 0.24. The indentation of each film was carried out about eight times and the data mostly close to the mean value was used.

### 2.6. Characterization of Wear Properties

The tribological tests were carried out through a ball-on-disc tribometer (UMT-2, CETR, America) at RT and 400 °C in ambient air. The tests were conducted against an Ø 5 mm Si_3_N_4_ ball at a normal load of 5 N for 15000 rotating cycles, at a speed of 500 rpm corresponding to 0.13 m/s.

The wear rates were determined by measuring the cross of wear track using the above-mentioned profilometer. The structure and composition of the film surface after wear the test were characterized by SEM and the attached EDX. Microstructure evolution was carried out by TEM observing the cross-section of the selected film.

## 3. Results and Discussion

### 3.1. Microstructure

Table 1 shows the applied Ag target power and some main structural parameters of the CrN_x_/Ag s. For better description, the samples were labelled as S1 ~ S5 according to the deposition parameters shown in the table. For reaching comparable film thickness, the number of periods decreased with the increasing Ag layer thickness, as shown in the table. The thicknesses of films range between 3.8 to 4.4 µm, including their Cr/Cr-N_x_ interlayer of about 0.2 µm. The thickness of the CrN_x_ top layer in the single layer was about 4.2 μm by 60 min deposition, which results in a growing rate of ~70 nm/min for CrN_x_. The multilayer films were prepared with a constant CrN_x_ sublayer thickness (*l*_CrNx_) of 60 nm, but with variant Ag layer thickness (*l*_Ag_) by adjusting the Ag target power density in different films. By measuring and calculating the average bilayer thickness, the mean Ag layer thicknesses for S2 to S5 were deduced from 3 to 10 nm, respectively. The EDS results revealed their Ag content increasing from 5.4 to 17.1 at.%, which correlates with the results of the Ag layer thickness ratio by calculating the volume fraction. The atomic Cr/N ratio of the CrN_x_ layer was kept at 2 ± 0.2 in all of the films.

Prior studies generally prepared nitride (carbide)/Ag multilayer films with compositional layer thicknesses less than 20 nm [3,20,21]. Such layer structures show enhanced hardness and/or toughness, however, it is relatively difficult to characterize the structure evolution behavior within a thin layer. The deposition parameters are tuned in this study to achieving a CrN_x_ layer thickness of 60 nm, which contributes to the study of layer structure evolution, especially the Ag out-diffusion behavior.

The XRD pattern of all the films is shown in Figure 1. The CrN_x_ single layer mainly shows one broad peak at 2θ = 44.162°. S2 is the multilayer film prepared with an Ag target power density of 0.65 W/cm^2^, in which another peak at 38.339° emerges. With the increase of Ag layer thickness, as shown in S3, S4 and S5, the XRD intensity at 38.339° and 44.162° continuously increases, while the diffraction peak at 38.339° gradually becomes dominant. Meanwhile, the diffraction peak at 64.877° and 77.931° emerges since S3 and their intensities increase with the increasing *l*_Ag_.

The possible phases of polycrystalline Cr, Cr_2_N, CrN, Ag, and Si are distinguished according to the JCPDS: 06-0694, 35-0803, 11-0065, 04-0783, and 27-1402, respectively. In addition, their standard peak positions are also illustrated in Figure 1. The broad peak at 2θ = 44.162° possibly results from the superposition of CrN (200) and Cr (110) diffraction peaks. A peak at 38.339° reveals that the Ag (111) diffraction emerges in S2, and the intensity increases with the increasing *l*_Ag_. Meanwhile, the diffraction peak at 44.162° gradually shifts towards Ag (200) diffraction angle and the intensity grows stronger with increasing *l*_Ag_, indicating that the sizes of Ag grain grow much larger than those of CrN_x_ and/or Cr. The diffraction peaks at 64.877° and 77.931° correspond respectively to Ag (220) and Ag (311). These results reveal well crystallization of Ag. It is frequently reported that the chrome nitride with a Cr/N ratio around 2 comprises Cr_2_N phases [27]. However, in this study, though all the films possess a Cr/N ratio near 2, the XRD Cr_2_N phase was not directly observed.

Figure 2 presents the cross-sectional microstructure of S2 by TEM observation. A general view of the layer structure shown in Figure 2a reveals that the 3-nm-thick Ag layers grow into un-continuous agglomerates (the deeper contrast pattern). As shown in Figure 2b, a high resolution view focused on Ag agglomerates and CrN_x_/Ag interface reveals that Ag preferentially grows into near-spherical granules (marked by the yellow arrows). The spacing d = 0.2355 nm correlates to Ag (111) according to PDF #65-2871. The observed size of Ag granules ranges between 5–10 nm.

Due to chemical immiscibility between Ag and CrN_x_ composition, the kinetic energy of the Ag species and surface energy of Ag atoms (or agglomerates) play important roles during the deposition and growth of the Ag layer, which results in the “island” growth mode of Ag grains on CrN_x_. The same result has been reported in [28]. However, as the average Ag layer thickness is 3 nm, it is insufficient to form a continuous layer in S2.

According to the high-resolution view of the CrN_x_ layer shown in Figure 2c, it is revealed that CrN_x_ possesses a nanocomposite microstructure comprising fine-sized crystallites and quasi-amorphous clusters. The measured planar spacing d = 0.2061 nm and 0.2038 nm coincides with standard cubic CrN (200) and Cr (110), respectively. By measuring the sizes of crystallites, the average grain sizes of CrN (200) and Cr (110) are 7.2 ± 2 and 9.5 ± 3 nm, respectively. The crystallite with spacing d = 0.2118 nm corresponding to Cr_2_N (111) is occasionally observed.

Figure 2d reveals the selected area electron diffraction (SAED) pattern of the multilayer structure as shown in Figure 2a. A broad and strong diffraction ring was found for the lattice spacing around 0.2038 nm, which correlates well with the XRD results. Weaker rings were discovered at spacings of 0.2343 nm, 0.1432 nm, 0.1173 nm, 0.1018 nm, and 0.0917 nm. The strong diffraction ring at spacing around 0.2038 nm probably results from a mixed diffraction of Cr (110), Ag (200) and CrN (200). These results corelate well with the XRD pattern of S2. The absence of Cr_2_N diffraction in both XRD and SAED patterns indicates a low volume fraction or small Cr_2_N grain size in the films.

The XRD and TEM results suggest that the grain sizes of CrN and Cr are quite small. In multilayer films, the grain size of Ag significantly increases with increasing *l*_Ag_, especially for the Ag grains with (111) plane paralleling to the layer surface. The grain sizes of Ag (111) and Ag (200), calculated by measuring the full width at half maximum of the diffraction peaks and TEM observation, are shown in Figure 3. The Ag (111) grain size is 9 ± 1.5 nm in S3 and 21.1 ± 3.2 nm in S5, while the (200) grain size is 5.2 ± 1 nm in S3 and 9.1 ± 1.6 nm in S5, respectively.

Such preferential growth of Ag (111) is ascribed to high free energy of Ag deposited on the CrN surface [29,30]. Since Ag is chemically immiscible with either Cr or N, the nucleation of f.c.c Ag tends to expose (111) with the lowest surface energy and gradually possesses preferential (111) grain growth.

It is also shown that the thickness uniformity (Figure 2a) and layer smoothness (Figure 2b) are well controlled. Though the Ag layer presents a coarse granulated surface, the top side of each CrN_x_ layer is quite smooth in a nano scale (Figure 2b). The smoother CrN_x_ layer surface is ascribed to the un-stoichiometrical Cr–N condensation that facilitates the deposition of CrN_x_ sublayers comprising small-sized Cr and CrN grains, which retards the preferential growth of either Cr or CrN, as well as their crystal orientations.

### 3.2. Mechanical Properties

The hardness and elastic modulus of films continuously decrease with the increasing *l*_Ag_. As shown in Figure 4, CrN_x_ single layer possesses a hardness of 24.3 ± 1.4 GPa, while that of S2 and S3 are 21.6 ± 1.8 GPa and 16.8 ± 2.1 GPa, respectively. The hardness of S1 is much higher than common CrN film (15 ~ 20 GPa) [31,32,33]. One major reason is the contribution of a nanocomposite microstructure composing of fine grains and quasi-amorphous clusters, which prevents the formation of a porous columnar microstructure and constrains the deformation by dislocation movement. Thus, S1 possesses enhanced resistance to plastic deformation, as proven by the elastic modulus as high as 332.6 ± 6.7 GPa. However, the multilayer design of CrN_x_/Ag results in lower hardness than the CrN_x_ single layer, and the hardness of S5 reduces to as low as 11.4 ± 0.5 GPa.

The insert shows the displacement of each film during the load-unload process. As the displacement of films was nearly the same, it is discovered that pure CrN_x_ film shows the largest stiffness and elastic recovery. The deformation resistance and elastic recovery reduce with the increasing *l*_Ag_, which results in a continuous decrease of hardness and elastic modulus. References of Ag-containing multilayer films have reported the same results [34].

There are a lot of factors affecting the hardness of multilayer films, including mainly, the rule of mixture and the interface effect [35,36]. In this study, the hardness and elastic modulus decrease by Ag incorporation, as well as decrease with the increasing *l*_Ag_. The hardness evolution indicates that due to low interface bonding strength between CrN_x_ and Ag, the bilayer interface provides few contributions to the hardness enhancement. On the contrary, the Ag layer promotes the plastic deformation of the multilayer. Baraket et al. reported similar results that by decreasing the thickness of nitride layers at a constant *l*_Ag_ of 4 nm, the hardness of multilayer films decreased [20].

The films are suffered to wear against Si_3_N_4_ ball for 30 min in 500 °C air, and the friction coefficient of each film is shown in Figure 5. The friction coefficient of the CrN_x_ single layer gradually increases within the beginning at 600 s, corresponding to a running in stage. The average friction coefficient of the CrN_x_ single layer at the stable wear process is 0.45.

Comparing with the CrN_x_ single layer, S2 provides a similar curve but a shorter running in stage. The friction coefficient at the stable wear process is 0.36. S3 presents the shortest running in stage and a very stale wear process, with the friction coefficient kept at 0.38±0.02.

The running in stage corresponds to the deformation and wear of films when they are exposed to a hard Si_3_N_4_ ball. The observed period of the running in stage reduces with the increasing *l*_Ag_. The possible reasons are increased surface deformability and/or faster solid lubrication performance, as *l*_Ag_ increases.

In S4 and S5, the friction coefficients at stale wear processes are as low as 0.26 and 0.2, respectively. However, the coefficient for S4 gradually rises after 22 min sliding, and that of S5 rises just from the 13th min. It is shown that the friction coefficient of films generally decreases with the increasing *l*_Ag_, indicating an effect of Ag lubrication to the sliding counterparts.

After sliding, the wear morphology of each film was revealed by measuring the surface depth across the wear track and profiled in Figure 6. The wear tracks generally go deeper and wider as *l*_Ag_ increases. The wear depth of S1, S2 and S3 are in the range between 2.5 to 3.2μm, while that of S4 and S5 reaches 5.5 and 8μm, respectively. The depth profile of the films suggests that S4 and S5 are worn through, while the wear loss is retained in S2 and S3. Correspondingly, the friction coefficient shifting to a higher value during sliding results from the exposure of the substrate in both S4 and S5.

Figure 7 shows the wear morphology of typical films using SEM characterization. The cross-sectional view reveals that the wear debris is adhered to in S3 and S4, but absent in S1, as shown in Figure 7a. The volume of debris generally increases with the increasing *l*_Ag_.

Plane views of the selected wear surface from S1, S3 and S4 are respectively shown in Figure 7b–d. Inserts in each film reveal the mass distribution of typical compositions. Furthermore, an arrow in red is marked on the wear track of each film and guides the sliding wear direction. S1, S3 and S4 reveal different wear morphologies. Figure 7b of S1 presents a smooth wear surface and loosely accumulated debris. As shown in the insert, increased Si concentration detected at the wear track zone reveals the wear loss of film, which is consistent with the result of the cross-sectional view and its wear track profile. The debris containing oxides of Cr and Si is loosely accumulated at the edge of the wear track. Thus, the debris is easily removed (by air-blowing), as shown in Figure 7a.

The sliding contact surface of S3 is covered by a flattened layer. EDS detection on the layer-covered wear surface reveals the presence of O, Cr and Ag, but the absence of N. The accumulated wear debris might be bound by Ag, unlike in S1, to become a protective layer through repeated sliding contact. As marked by an open circle in Figure 7c, the adhesion of the layer to the film surface might be poor, however, the presence of a surface-covering layer contributes to the reduction of friction and wear.

In S4 and S5, wear debris accumulated at the edges of wear track instead of cover the wear surface, which results in severe wear of both films. As this result is contradictory to the statement in S3, further understanding on the mechanism of solid lubrication as well as Ag migration process is necessary.

The high temperature tribological behavior of Ceramic/Ag films, both nanocomposite and nano-multilayer (periods within 20 nm), have been widely studied [9,10,11,37,38,39]. It is well recognized that the out-diffusion of Ag to the film surface provides a solid lubrication effect, and there exists an optimal Ag concentration and/or *l*_Ag_ to achieve low friction and wear [2,3,5,20]. However, for the present CrN_x_/Ag multilayer films, how does Ag transport through the 60-nm-thick CrN_x_ barrier?

During the process of TEM observing the microstructure of the thin CrN_x_/Ag multilayer film, the grain morphology of Ag changes. It gradually become rounder or larger in size, leaving voids at the CrN_x_/Ag interface, as shown in Figure 2a. Chiodi et al. have reported out-diffusion of Ag due to high surface energy and internal stress of Ag grains in a CrN/Ag multilayer film, while the activation energy is provided by heating or even reduced by the atmosphere [40]. In this study, electron radiation during TEM observation results in the microstructure evolution.

Mulligan et al. studied the behavior of Ag diffusion in CrN/Ag nanocomposite films and reported that it is a detachment limited process [11]. In S2, the isolated Ag agglomerates were embedded in an CrN_x_ layers/matrix, showing the same mechanism of Ag transportation as in the CrN/Ag nanocomposite films. This process is supported by the presence of large amounts of voids and cracks in S2, as shown in Figure 2a.

The voids locate at the CrN_x_/Ag interface or displace preferentially at Ag sites. It is regarded as a result of Ag recrystallization or detachment of Ag grains. The cracks may play an important role in Ag transportation. As Ag tends to detach rather than diffuse to the surface at ambient temperature, cracks are probably the channels or remains of Ag migration.

Due to low Ag concentration, S2 shows a limited effect of Ag diffusion on its friction and wear properties. For films with higher Ag concentration, it is worth regarding the microstructure evolution as a key factor affecting the tribology of films.

The cross-sectional microstructure of S3 before and after 500 °C sliding was examined, as shown in Figure 8. The as-deposited microstructure of S3, as shown in Figure 8a, reveals a fine layered structure with equal-thick Ag layers. By observing the microstructure of S3 after a tribological test, cracks and layer distortions are shown in Figure 8b.

The above characterizations show that the microstructure of CrN_x_ layers are dense and finely crystallized. Therefore, the presence of cracks in CrN_x_ layers, both for S2 and S3, tend to generate during the post-deposition test or characterization. There are two possible reasons for the formation of cracks. Firstly, the process of Ag inter-layer accumulation results in stress, deformation and even cracks in the CrN_x_ layer. The Ag accumulation process can be triggered at an environmental temperature higher than it is deposited, or by heat radiation such as an electron beam. Secondly, during the high temperature sliding process, repeated contact loading on the film’s surface results in a stress and strain field in the films, and it probably contributes to the generation of cracks. The size of cracks in S3, as shown in Figure 8b, is larger than in S2, so are the layer distortions.

The open square in Figure 8b marks one of the Ag accumulation structures, who presents a larger size than common *l*_Ag_. A magnified view of the marked image is shown in Figure 8c. It reveals that not only a large-sized Ag agglomerate is formed, but also *l*_Ag_ grows to as large as 15.2 nm.

The inset of Figure 8c shows a fast Fourier transform pattern corresponding to the microstructure marked by an open square, which reveals the agglomerate as a single-crystal Ag. Abnormal growth of *l*_Ag_ and agglomerates indicates that S3 has been through a severe process of Ag accumulation and out-migration.

During high temperature sliding, the friction coefficient at the stable wear process decreases with the increasing *l*_Ag_, due to more amount of Ag migration in the larger *l*_Ag_ films. The length of the running in stage also reduces with the increasing *l*_Ag_, as a result of faster Ag migration in the larger *l*_Ag_ films.

With the out-migration of Ag, cracks, voids and layer distortions emerge in the films, which make the films become loose and the strength deteriorated. The wear resistance of films strongly depends on the hardness and load bearing capacity. S4 and S5 present low friction coefficients but high wear loss, probably due to poor structural compactness and load bearing capacity, as a result of serious Ag migration.

Great interests have been focused on solid lubrication films with long term wear resistance. This study shows that optimal *l*_Ag_ lies at about 4.5 nm. However, optimal Ag content depends largely on the film structure, deposition tool and service condition. Driven by the free energy, the process of Ag out-migration is inevitable and continuous if the channel exists. C.P. Mulligan suggested to tune the mass of Ag transport through a CrN cap layer as thick as 200 nm [13]. This study also reveals that 60-nm-dense CrN_x_ sublayers can hardly obstruct the out-migration of Ag.

In Ceramic/Ag composite and/or multilayer films, the free energy of Ag plays an important role in deciding the structure evolution process. Beside the barrier layers, tuning the microstructure and energy of Ag composition may offer another route to valve the mass-migration.

## 4. Conclusions

The present study focuses on the microstructure and mechanical properties of CrN_x_/Ag multilayer films prepared via reactive magnetron sputtering. The multilayer films have a constant *l*_CrNx_ of 60 nm, while *l*_Ag_ ranges from 3 to 10 nm. The results are listed as:

(1) The microstructure of CrN_x_ layer comprises Cr and CrN. The multilayer films with *l*_Ag_ lower than 4.5 nm show isolated Ag grains embedded in CrN_x_ layers;

(2) CrN_x_ single layer possesses a hardness of 24.3 ± 1.4 GPa, while it reduces constantly with the increasing *l*_Ag_. Wear test in 500 °C air reveals that the friction coefficient decreases with the increasing *l*_Ag_. The *l*_Ag_ = 6 nm film (S4) was worn through and serious wear lost was revealed for *l*_Ag_ = 10 nm (S5);

(3) CrN_x_ single layer (S1) and the *l*_Ag_ = 3 nm film (S2) show low wear rate probably due to high hardness, while a protective layer on the surface of the *l*_Ag_ = 4.5 nm film (S3), formed by accumulated wear debris, reduced the wear and friction.

## Figures and Tables

**Figure 1 materials-13-01316-f001:**
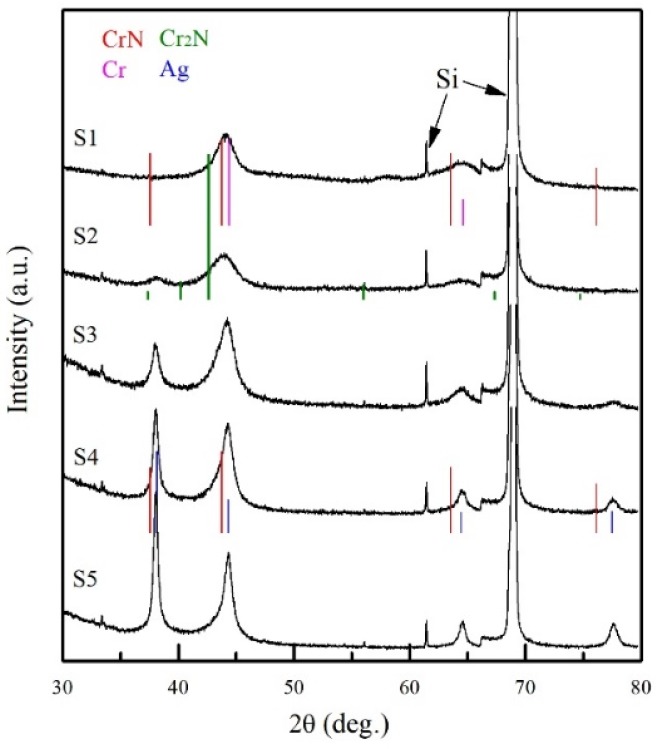
XRD pattern of CrN_x_ single layer and CrN_x_/Ag multilayer films.

**Figure 2 materials-13-01316-f002:**
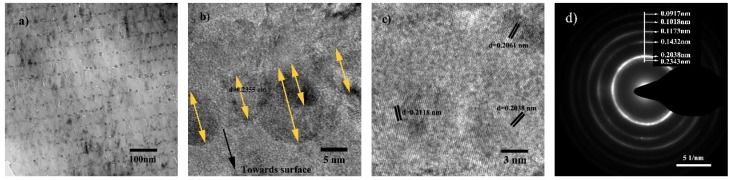
Cross-sectional TEM image of S2: (**a**) general view; (**b**) high resolution view focused on Ag agglomerates and CrN_x_/Ag interface; (**c**) high resolution view focused on CrN_x_ sublayer; (**d**) selected area electron diffraction pattern of the multilayer.

**Figure 3 materials-13-01316-f003:**
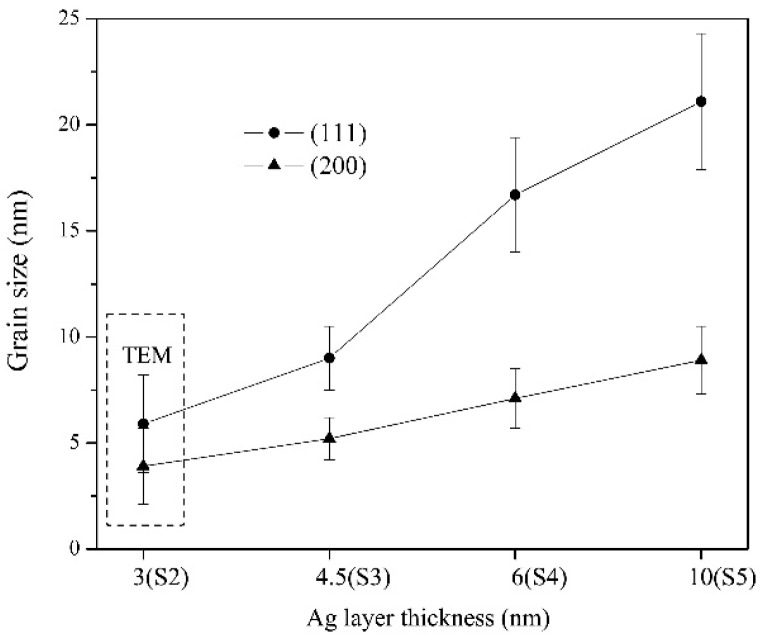
Plot of the variation of Ag (111) and Ag (200) grain size against Ag layer thickness.

**Figure 4 materials-13-01316-f004:**
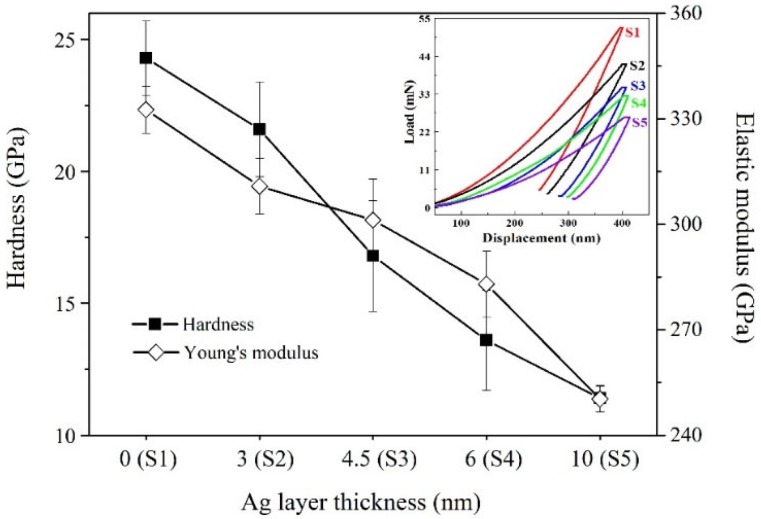
Hardness and elastic modulus of multilayer films at different Ag layer thicknesses. The insert shows the typical load-displacement curves of the studied samples.

**Figure 5 materials-13-01316-f005:**
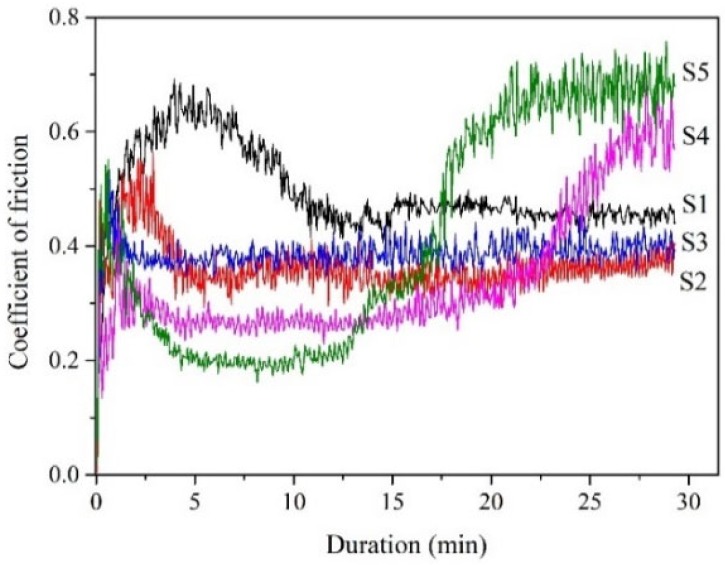
Curves of friction coefficients of the films sliding against Si_3_N_4_ ball in 500 °C air.

**Figure 6 materials-13-01316-f006:**
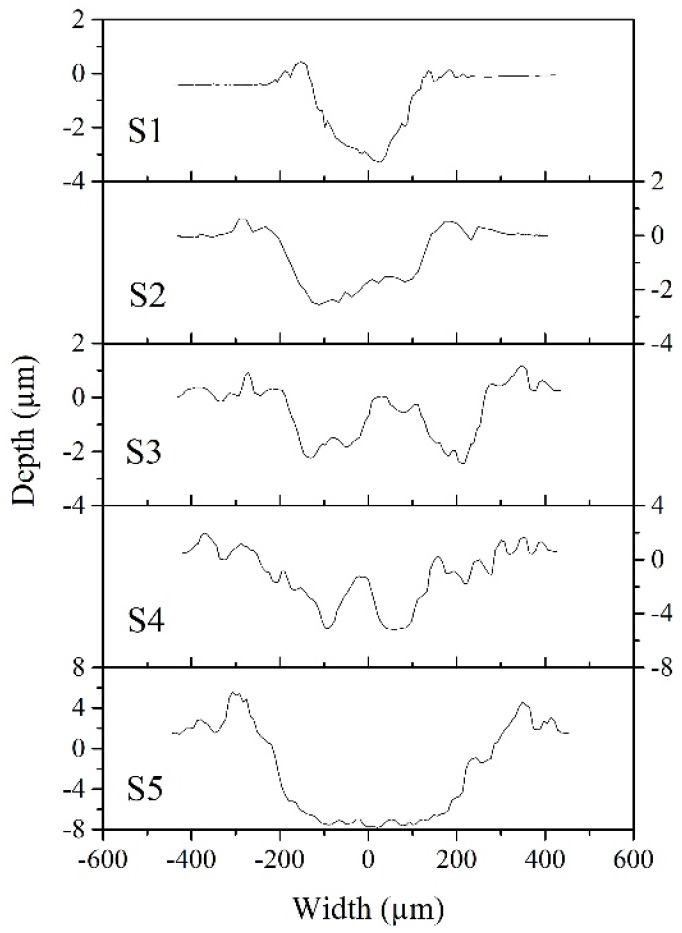
Depth profiles across the wear tracks of films.

**Figure 7 materials-13-01316-f007:**
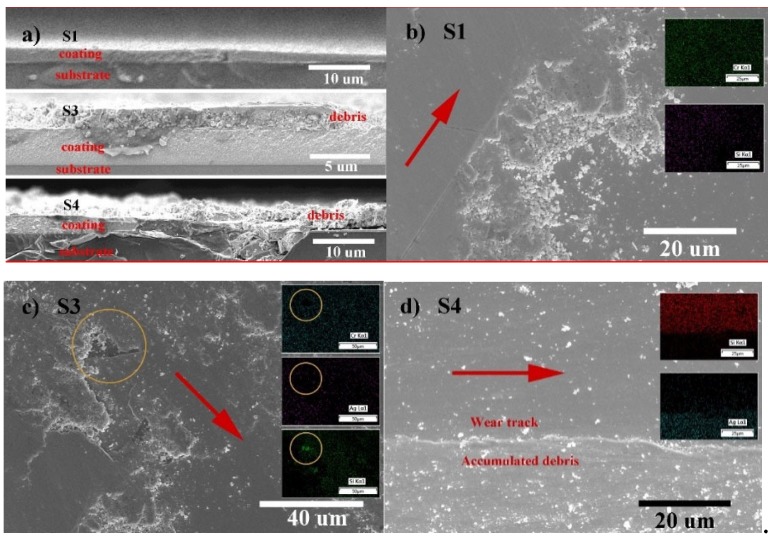
Wear morphology of typical films characterized through a planar and cross-sectional view. (**a**) Cross-sectional view focused on the wear track and debris of selected films; (**b**) plane view of S1, inserts show the mass distribution of Si and Cr; (**c**) plane view of S3, inserts show the mass distribution of Cr, Si and Ag; and (**d**) plane view of S4, insert shows the mass distribution of Si and Ag.

**Figure 8 materials-13-01316-f008:**
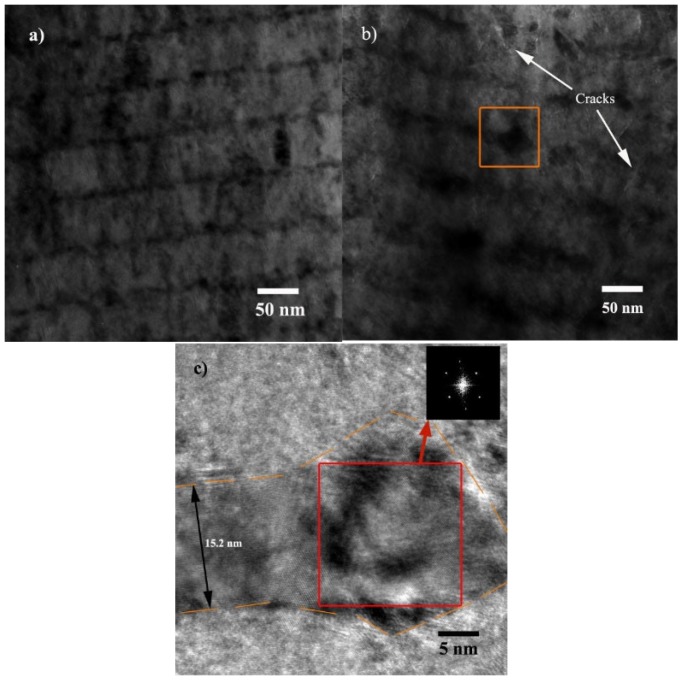
Cross-sectional microstructure of S3. (**a**) As-deposited; (**b**) after 500 °C sliding microstructure; and (**c**) magnification of the open square marked view shown in (**b**).

**Table 1 materials-13-01316-t001:** Main structural parameters of deposited coatings.

Sample	Ag Target Power Density (W/cm^2^)	Average Layer Thickness (nm)		Periods Number	Film Thickness (µm)	Film Content (at.%)		
		Ag	CrN_x_			Cr	N	Ag
S1	0	0	–	–	4.4 ± 0.3	67.2	32.8	0
S2	0.65	3	60	65	4.1 ± 0.2	62.1	32.5	5.4
S3	1	4.5	60	60	4 ± 0.2	61.7	31.2	7.1
S4	1.3	6	60	55	3.8 ± 0.2	58.6	30.6	10.8
S5	2	10	60	55	4 ± 0.2	54.2	28.7	17.1

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
