# Peer review of "On the Microstructure and Mechanical Properties of CrN_x_/Ag Multilayer Films Prepared by Magnetron Sputtering"

_materials, 2020, doi:10.3390/ma13061316_

Round 1

Reviewer 1 Report

Awkwardly written in line 23: “with increasing the Ag layer thickness, 23 the hardness ever decreases from 30.2 to 11.6 GPa.” Confirm the use of the word “ever” here in this context.

The authors failed to explained and discussed the parameters: “repetition” in table 1

Awkwardly written: “The XRD patterns of S3, S4 and S5 show ever 120 increased diffraction intensity at both 38.339° and 44.162°”

Figure 1 should have better spacing in height between each sample XRD profile result.

Line 142 to 145 : “Due to chemical immiscibility between Ag and CrNx composition, the kinetic energy of the Ag species and surface energy of Ag atoms (or agglomerates) play important roles during the deposition and growth of Ag layer, which results in the “island” growth mode of Ag grains on CrNx. However,as the average Ag layer thickness is 3 nm, it’s insufficient to form a continuous layer in S2.” What a continuous layer formed for higher Ag layer thickness, so that you can infer the conclusion above?

Figure 2 should have figure a,b,c and d spaced between them, or with a fine contour.

Line 165 and 166 the authors should reference the Scherrer equation of the method used to measure grain size. Moreover, the authors should mention what was the function or the refinement method used to approximate the experimental XRD curves. Examples: Pearson VII, Voigt model?

Figure 3: In the legend should be written: Plot of the variation…

Line 171 : "Such preferential growth of Ag (111) is ascribed to high free energy of Ag deposited on CrN surface". Please discussed or show a reference.

Figure 4 .” Insert shows the typical load-displacement curves of the studied samples”. What is the purpose of the load-displacement here? And where were these results taken from?

Author Response

Dear reviewer, thank you for your comments, please find our response in the attachment.

Reviewer 2 Report

Please find attached a PDF file with my comments and suggestions for authors.

Author Response

(The authors gave the same response as above.)

Reviewer 3 Report

Dear Author,

I have read the manuscript "On the microstructure and mechanical properties of CrNx/Ag multilayer films" and I have listed my suggestions below.

General remarks:

I propost to change the title. In my opinion your title did not describe the content of the paper. As you wrote in the Conclusion "The present study focuses on the microstructure and mechanical properties of CrNx/Ag 292 multilayer films prepared via reactive magnetron sputtering." I propose to add info about reactive magnetron sputtering to title. In my opinion the list of references is poor. You have presented 28 references. The two newest have been published in 2016 and 2012. I am strongly suggesting to add positions published in last couple of years. There are a lot of articles adequate to yout experiments.

Introduction:

You have started this paragraph from "In recent years". However, you did not presented any of latest references. Please add information from latest published articles, which are related to content of your manuscript (eg. magnetron sputtering, multilayer films). Please clearly indicate the novelty of your work. You have not presented the significant background of your work, so it is very hard to see the novelty.

Experimental:

It is not clear, why have you chosen these values of process parameters. From initial research, previous research or references? I propose to change the name of this paragraph to "Methodology".

Results:

You have presented the thickness of the films. How many measurements have you prepared for each calculation? I suggest to extend your discussion. You shoud compare your results with the results of other Researchers. How mamy specimens heve been tested in each test? You have presented the results, but it is not clear from how many specimens they have been caculated. Line 248 - avoid "it's" in scientific paper. Fig. 8. - We can see cracks in this figure. Pleas describe potential reasons. Why "These results indicate more and faster mass transportation of Ag toward film surface, as lAg grow thicker." Please describe the mechanism.

Conclusions:

This is the strongest part of your article. Conclusions are strongly connected with the results.

Author Response

Thank you very much for your time and kindly comments!

Please see the attachment。

Reviewer 4 Report

Nanoscale design of materials is a promising research path as these materials are needed ofr a wide range of applications. Reduction of the friction coefficient between moving parts is an active research field and several techniques have been developed for years.

The present manuscript focuses on the microstructure and mechanical properties of CrNx/Ag multilayer films regarding their interest in reducing friction. This type of films have already been studied in the literature but the authors provide some new elements concerning the influence of the thickness of the Ag layer and the interface microstructure on the various properties of the designed films. The manuscript is well structured and concise. The first part, the introduction provides a quick state of the art about CrN/Ag thin films. The second part presents the experimental setup and the techniques used in the study. The third part exposes the results, e.g. microsturture and mechanical properties. A conclusion briefly summarizes the main outcomes, some research perspectives would be welcomed though.

Besides completing the conclusion, the authors should correct some minor details:

L 87: Scherrer with capital S

l 96: Poisson with capital P

l 96: only 8 indentation points on a film? Can you explain this choice

Fig 1: please provide a larger figure as we cannot see details (is it the indicative Ag bar at 38°?)

l 198: maybe you should expose briefly the novelty regarding ref 17?

l 213: precise how does the depth profile is measured (instrument, etc)?

L 299: capital W in Wear

Author Response

(The authors gave the same response as above.)

Round 2

Reviewer 2 Report

An important part of my original comments have not been addressed by the authors. I note them next:

- Comment: In relation to the section “2 Experimental”, in my opinion it must be extended, explaining with more detail the experimental setup performed in the research included in the manuscript. I suggest to divide it in several subsections, one for explaining each experimental technique.

This comment has not be addressed. I have observed minimum changes in the revised version of the manuscript. It has been included two general subsections, but in my comment I suggested in divide it in several subsections, one for each one of the experimental techniques. In the current version of the manuscript, all the techniques descriptions are "mixed" in the subsection 2.2, making more difficult the understanding of the experimental setup performed. Please, include one subsection for each technique used.

- Comment: Regarding the results and discussion section, I suggest to improve it, especially the discussion of the results, which in my opinion is the most important part of the manuscript, because here the majority of this  section is dedicated to describe the results. Therefore, the discussion of results must be extended, discussing the results with more depth and detail. I also suggest to include more references in the discussion, because there are a scarce number of references cited in that section, and more references are necessary to support the discussion of the results obtained.

The changes in this section are residual (7 lines), and the discussion of results has not been improved. Only 2 references have been added, which are totally insufficient. Please, addressed this comment correctly.

- Comment: Regarding the conclusion section, I suggest to summarize the conclusions using bullet points or numbers in order to emphasize the most important findings of the manuscript. This can make the conclusions clearer.

It has been included the bullet points, despite that, I think that the conclusion still needs to be a bit more summarized, because it is excessively long which makes difficult to see the most important findings of the manuscript. 

Finally, I think that the authors should consider more seriously the comments of the reviewers, because my sensation is that it has not happened in the first review round. 

Reviewer 3 Report

Dear Authors,

I have read your manuscript very carefuly. You have included suggestions marked by reviewers. Your responseto my review is correct. However, I propose minor revision:

  • From your response:
    - 2) Some of the parameters were preferentially chosen. For example, previous studies generally prepared multilayer films with comparable thickness of alternating layer, or with layer thickness less than 20 nm. In this study, the applied power (correlated with target diameter) of targets and sputtering time resulted in a thickness of CrNx layer benefit for studying the diffusion of Ag or barrier effect."
    Pleas add clear info to the text. The response is ok, but I can not find this in revised manuscript.
  • line 207 - add space before bracket [ ],
  • From your response: "

    5) About cracks in Fig. 8-b, there are two possible reasons. First is that the process of Ag inter-layer accumulation results in the stress, deformation and even cracks in CrNx layer; second is that the stress and strain field in the film due to repeated sliding contact also results in or contributes to the generation of cracks."

    6) About the statement of "These results indicate more and faster mass transportation of Ag toward film surface, as lAg grow thicker", which mainly based on the comparing studies of friction coefficient, wear morphology and TEM microstructure. The out-diffusion of Ag is a free energy driven process. References have suggested that Ag moves through detachment rather than diffusion, while report concerning the migration path is few. This study suggests Ag migrates through the cross layer cracks of CrNX. Reasons for the emergence of cracks have been described, we will further study the factors that affecting the migration behaviour.

  •  These two responses are clear or me and I agree with them. However, the text still miss this valuable information. Please add this into the manuscript.

Round 3

Reviewer 2 Report

My comments and suggestions have been fulfilled, therefore I recommend to accept the manuscript.